# Fatty Acids, Volatile and Sensory Profile of Multigrain Biscuits Enriched with Spent Malt Rootles

**DOI:** 10.3390/molecules25030442

**Published:** 2020-01-21

**Authors:** Maria Simona Chiş, Anamaria Pop, Adriana Păucean, Sonia Ancuța Socaci, Ersilia Alexa, Simona Maria Man, Monica Bota, Sevastiţa Muste

**Affiliations:** 1Department of Food Engineering, Faculty of Food Science and Technology, University of Agricultural Sciences and Veterinary Medicine of Cluj-Napoca, 3–5, Manastur Street, 400372 Cluj-Napoca, Romania; simona.chis@usamvcluj.ro (M.S.C.); adriana.paucean@usamvcluj.ro (A.P.); simona.man@usamvcluj.ro (S.M.M.); monica.bota@usamvcluj.ro (M.B.); sevastita.muste@usamvcluj.ro (S.M.); 2Department of Food Science, Faculty of Food Science and Technology, University of Agricultural Sciences and Veterinary Medicine of Cluj-Napoca, 3–5, Manastur Street, 400372 Cluj-Napoca, Romania; sonia.socaci@usamvcluj.ro; 3Department of Food Control, Faculty of Agro-food Technologies, Banat’s University of Agricultural Sciences and Veterinary Medicine “King Michael I of Romania”, 119 Calea Aradului Street, 300641 Timişoara, Romania; alexa.ersilia@yahoo.ro

**Keywords:** biscuits, malt spent rootlets, fatty acids, volatiles, food waste

## Abstract

Spent malt rootlets, a by-product of the brewing industry, are a rich source of protein, essential amino acids, healthy fats, polyphenols and minerals, and could be a new promising type of raw material from the nutritional, economic, sensory, and technical perspectives. However, their specific aroma profile could limit their addition in baked products. The aim of this work was to study the effect of spent malt rootlets addition on volatile derivatives of enriched biscuits in relation to their sensory profile. For this purpose, spent malt rootlets and enriched biscuits (0–25% spent malt rootlets added) were analyzed by GC-MS techniques, in order to obtain their fatty acids methyl esters and volatile compounds profile, while for the sensory analysis a nine-point hedonic score test was used. The results of this study reveal the fatty acids and volatile profile of spent malt rootlets and of the enriched biscuits with spent malt rootlets pointing out the contribution of fatty acids to the generation of aroma compounds. The influence of different aroma compounds on the consumer’s preferences was studied and the optimum level addition of spent malt rootlets in multigrain biscuits was found to be 15%.

## 1. Introduction

Food waste, one of the largest portions of solid waste, has grown to become a global problem. Every year, nearly 1.3 billion tons of food trash is dumped in landfills or is otherwise disposed of [1]. Its efficient management represents the main objectives of European Union (EU) actions against food waste and towards sustainable development, aiming to halve food waste per capita at the retail and consumer level by 2030, and reduce food loss among the food production and supply chains [2]. Nowadays, efforts are being directed towards the exploitation of agro-industrial waste, from both economic and environmental standpoints. These by-products are recognized as having significant amounts of biologically active compounds, namely proteins, fibers, polysaccharides, flavor compounds, or different phytochemicals, while their uncontrolled disposals lead to environmental problems [3,4]. These bioactive compounds can be revalorized as functional ingredients in food, pharmaceutical, health care, cosmetic, and other products. Using the recovered bioactive molecules as functional ingredients represents a sustainable alternative of food wastes exploitation as an inexpensive source of valuable compounds, by developing innovative food and non-food products with health-promoting benefits and at the same time contributing to an efficient waste reduction management [5].

Barley (*Hordeum vulgare*), a cereal of the Poaceae family, ranks fourth globally in both quantity produced and cultivation area. It is conventionally utilized for animal fodder (≈66%) or malted for brewing (≈33%) and food applications (≈2%) [6]. During malting, the sprouted green malt grains are ‘cleaned’, by removing the protein-rich rootlets, which are primarily discarded or used as animal fodder. Recently, the incorporation of barley into the human diet has received renewed attention due to increased scientific evidence showing it to be an excellent source of dietary fiber, particularly β-glucan [7]. β-Glucan is a soluble fiber that has health benefits, which include decreasing the risk of heart disease and being effective in lowering blood serum glucose levels [8,9].

Barley malt rootlets, a by-product of the brewing industry, consist of the plumule and radicle of barley and may include some of the malt hulls [10]. Barley rootlets could contribute to the human diet through their valuable nutrients such as protein, essential amino acids, healthy fats, polyphenols, and minerals [7]. Their protein levels are far in excess of those detected in barley or malt grains 31.90–36.75% [7,11]. Waters et al. [7] reported that the ten essential amino acids are heavily represented in the rootlet portion of malt and are in excess of the wheat flour, whole meal flour, barley and malt levels for phenylalanine, isoleucine and leucine.

Capitalizing on these agro-industrial wastes into value-added products has recently become a highly active area of research [12]. Thus, the incorporation, of rootlets into a food product could increase the protein content of poorer flours and thus increase the net nutritive value of the final baked goods. Rootlets are considered as an inexpensive source for enriching food due to its composition on high biological protein value, being first used as a raw material in bakery products in 1997 [13].

The literature survey has clearly shown that spent malt rootlets belongs to the GRAS (generaly recognised as safe) category and could be a new promising raw material from the nutritional, economic, sensory and technical perspectives [14,15,16,17].

However, changing ingredients may cause changes in reaction precursors and thus may have an impact on the main reactions occurring during the steps of product-making, such as the Maillard reaction, caramelization, and lipid oxidation [18].

In this context, this research aimed to study the effect of spent malt rootlets addition on volatile derivatives of enriched biscuits in relation to their sensory profile. The critical issue in the case of the addition of spent malt rootlets in food is their sensory profile so it is important to establish the optimum addition percentage through the sensory analysis in order to correlate the results of instrumental analysis to human perception. A high quality mix of multigrain (spelt wheat, rice, and oat), pseudo-cereal (buckwheat), and chia seeds was used in biscuit formulation in order to improve various nutritional parameters.

## 2. Results and Discussion

### 2.1. The Proximate Composition of the Main Raw Materials

The proximate composition of the main raw materials used for the manufacture of the biscuits is showed in Table 1. The protein and ash contents of SWF (15% and 1.25%, respectively) are close to the results reported by [19], as well as [20] for whole spelt flour, namely ash (1.95%), protein (15.17%), and moisture (15.04%). The lipids content is similar with the amount reported by [21] ranging between 3.24–3.5%. The small differences between the results could be due to the environmental factors, climate [22] as well as due to milling, storage, and manipulation. The proximate composition of buckwheat, rice flours and oat flakes is similar to that reported by [23], being good sources of proteins, lipids and minerals for human diet. Due to their nutritive value these flours are often selected as raw materials for nutritionally balanced biscuits.

Regarding the MRF, it’s proximate composition was in the typical ranges as it was reported by literature; the protein content of 35.5% and the total lipid amount (1.9%) are close to the values reported by [7]. The carbohydrates content reached a similar value (46%) to that reported by [24], while the ash content was similar with the amount reported by [25].

It has been reported that normal biscuits are nutritionally deficient as they are low in protein, dietary fiber, as well as vitamins and minerals. Several studies have reported on improving the nutritional quality of biscuits by using protein rich ingredients like oil seed meals [26], pulses [27], and industrial by-products like wheat germ [28]. Recently, a few studies have reported the use of multigrain to improve the nutritional quality of bread and other traditional products [29]. These results could sustain the selection of these raw materials, including the spent malt rootlets as a better choice for producing baked goods with enhanced nutritional value. Further papers will report on the techno-functional properties of the enriched biscuits with MRF.

### 2.2. Fatty Acids Methyl Esters Content of Spent Malt Rootlets Flour (MRF) and Whole Wheat Spelt Flour (SWF)

In the present study, by using GS-MS analysis, a total number of 19 fatty acid methyl esters (FAMEs) were identified, while 16 of them where identified in MRF samples and only 13 in the SWF samples, as reported in Table 2. Eight saturated fatty acids (SFAs) were found, with palmitic acid being the predominant one in both analyzed flours but with an increased concentration for spent malt rootlets flour. Regarding monounsaturated fatty acids (MUFAs), seven compounds were found, among which, oleic acid was the major one. The content of oleic acid was identified as double in the case of whole spelt flour, around of 24.66%, comparing to MRF. Concerning polyunsaturated fatty acids (PUFAs), four compounds were found, standing out both linoleic and linolenic acids as major compounds. Spent malt rootlets flour recorded high values of linoleic and linolenic acids (35.617% and 32.64%, respectively). In the case of whole spelt flour only linoleic acid reached 26.696%, while linolenic acid content was 4.426%. Similar results were reported by [7], showing that linoleic, linolenic, palmitic and oleic are the main fatty acids from spent malt rootlets.

Regarding the lipid content of spelt wheat flour [30], reported that compared to the common wheat, the lipid’s spelt content is higher and varies between 7–30% and palmitic acid, oleic acid, linoleic acid and linolenic acid are the main fatty acids from spelt wheat.

By these results, MRF show a good fatty acids profile with high amount of PUFAs (ω−6 and ω−3), essential fatty acids that must be derived from the diet, cannot be made by humans, and other mammals mainly because of the lack of endogenous enzymes such as delta 12 and delta 15, accordingly to [31] and [32].

However, the presence of fatty acids in MRF contributes also to the generation of volatiles in biscuits during the manufacturing process. Linoleic and linolenic acids are usually oxidized by lipoxygenase action, resulting hydroperoxides which are unstable and are gradated during baking, mainly into hexanal and hexenal. Lipoxygenase is more active in dough without yeast, more oxygen remain disposable for this enzyme to transform lipids into aldehydes, ketones, and esters [33].

Out of the total of fatty acids, the high content of ∑PUFAs in MRF is noticeable in a ratio of 4.95, compared to 1.22 in the SWF, this proves the positive effect for health.

### 2.3. Volatile Compounds of Spent Malt Rootlets Flour (MRF) and Whole Wheat Spelt Flour (SWF)

In order to achieve this goal, a total number of 21 aroma compounds were analyzed by using ITEX/GS-MS, as shown in Table 3, which were divided into five classes for an easier discussion: alcohols, aldehydes, ketones, terpenes and terpenoids, others.

Compared to whole wheat spelt flour, a greater number of volatile compounds were separated in MRF samples. In MRF, a total number of 18 volatiles were identified, while SWF had only 4 identified volatile compounds. The main MRF identified aroma compounds from the alcohols group were 3-methylbutan-1-ol (40.21%), 2-methylbutan-1-ol (4.02%), and pentan-1-ol (3.25%), while for SWF, no compound from the alcohol class was identified.

The technological steps in processing barley into malt, such as steeping, germination and drying of the germinated seeds involve several reactions which are responsible for the generation of an enhanced aroma profile [34,35]. The germination temperature has a strong influence on the content of volatile compounds. The high amount of 3-methylbutan-1-ol, 2-methylpentan-3-one is generally correlated with germination temperature and green malt moisture content, being independent on the time germination, according to [36].

Hexanal (8.36%) was the predominant compound of the aldehydes group. From the ketone class in MRF, compounds like heptan-2-one, ethenone, 2-methylpentan-3-one and octan-3-one were identified, while in WSF only 1-phenylethanone (8.02%) was identified out of the ketone group. 1-phenylethanone is characterized by [37] as having a sweet, floral and almond aroma. Limonene (6.08%) and *p*-cymene were the volatile compounds from terpenes and terpenoids group, identified in MRF sample. 2-pentylfuran (10.25%) was the only volatile compound found in MRF representing the furans group, characterized by [38] as having an odor perception of green bean, butter. The aroma profile of spent malt rootlets is showed in Figure 1, pointing out that alcohols contributed mainly by giving whiskey, malt and alcoholic flavor notes as the main perceived aroma.

Whole spelt flour had as aldehydes compounds 2-methylpropanal (49.20%), hexanal (35.96%) and nonane (6.80%), probably derived by lipoxygenase action during milling and maturation processes.

### 2.4. Fatty Acids Methyl Esters Content of Final Enriched Biscuits

In the enriched biscuits, a total of 12 fatty acids methyl esters (FAMEs) were identified, from which seven were classified as being saturated (SFAs), three as MUFAs, and two as PUFAs. The main compounds from the saturated group were myristic acid reaching around of 11% of total fatty acids, followed by lauric acid and capric acid. From PUFAs acids, linoleic acid reached the highest value increasing significantly with the MRF addition in biscuits. Also, linolenic acid and myristoleic reached higher contents as the MRF content increased in biscuits formulation. As could be seen by comparing the fatty acids profiles (Table 2 and Table 4), linoleic acid was identified in a high amount in the control sample too, highlighting that spelt wheat flour also contributed to this content.

Fat is a key component in aroma perception being involved also in the texture of the final baked goods and improving the sensory features such as the mouth-feel [39]. Lipid oxidation is involved in the final characteristics of the food, such as aroma, taste, nutritional value, color, and texture [40]. According to [41], alcohols are formed mainly from the lipids through enzymatic reaction catalyzed by lipoxidase. Also, by auto-oxidation of unsaturated fatty acids (C18:1, C18:2) short chain aldehyde (C6-C9) could be formed [18]. Hexanal is a typical compound of linoleic and arachidonic acids oxidation. It is often used as a marker of lipid oxidation.

The ratio of fatty acids ∑PUFAs/∑MUFAs on MRF/SWF, Table 2, are correlated with enrich biscuits, as can be seen in Table 4, the ratio ∑PUFAs/∑MUFAs on B0 is 11.532, compared to 17.24 in B25, this ratio increases with the addition of MRF. According to [42], it seems that the PUFA/SFA ratio alone is not suitable to predict the changes of plasma lipids level and that the PUFA + MUFA/SFA ratio is a more suitable parameter for this purpose. Moreover, the authors [42] suggest that the main prerequisites for keeping low plasma and liver cholesterol are: (i) low MUFA/SFA ratio, (ii) high PUFA/MUFA ratio, and (iii) PUFA + MUFA/SFA ratio not to exceed 2. From this point of view, the sample B0, B5, B10 and B15 are meeting the above-mentioned criteria, as can be seen in Table 4.

### 2.5. Volatile Compounds Content in Enriched Biscuits

A total number of 18 volatile compounds were identified in the enriched biscuits, by means of ITEX/GC-MS technique as shown in Table 5. Also, the perceived aroma was correlated with the literature data, the perceived flavor of each identified compound is showed in the same Table 5 [37,43,44].

Limonene, *β*-pinene, hexanal, 3-methylbutan-1-ol were the principal volatile compounds found in the final baked biscuits, influencing the taste by having a citrus, mint flavor together with fresh, green, fruity, sweet or pine, resine and turpentine flavours (Figure 2). Limonene has been reported having chemo- preventive activity against different types of cancer such as breast or colorectal ones at a concentration between of 0.5–12 g/m^2^/day [45]. The amount of limonene could be correlated with the carotenoids amount, according to [41]. The high amount of limonene could be due to the high content in carotenoids of chia seeds used in biscuits formulation, since it is present even in the control sample in relative high amount 24.76% and the limonene content increased progressively with the MRF addition.

From the ketone group, 1-phenylethanone was the main volatile compound identified, being characterized by having an almond, floral flavor.

Hexanal, according to [38], is the end product of the lipoxygenases and hydroperoxide isomerases which are involved in the oxidation of unsaturated fatty acids. Also [46], mentioned that hexanal is the main product of linoleic acid autoxidation. This idea is supported by other authors [41] who mentioned that unsaturated fatty acids could influence the amount of aldehydes and ketones in flours.

*β*-pinene with pine, turpentine and traces of mint, camphor and eucalyptus aromas and *α*-pinene with a citrus and spicy, woody pine aroma contribute significantly to the aroma profile of enriched biscuits (Figure 2). *α* and *β*-pinene are important compounds used in food preservation, due to their antimicrobial activity. More than that, *α*-pinene has been reported to have antitumor activity on a concentration of 8 mg/L [47].

The presence of 3-methylbutan-1-ol in enriched biscuits could be explained by the spent malt rootlets addition since this compound was not identified in the control sample and MRF contained high amounts of this compound (Table 3).

Undoubtedly, the other ingredients from the biscuits formulation (multigrain mix, butter, malt extract) also contributed to the final volatile derivatives content and to the biscuits aroma profile through lipid oxidation, Maillard reactions, including Strecker degradation of the carbonyl compounds and caramelization [33]. These ingredients act as a possible source of precursors for volatile compounds due to their chemical composition of lipids and antioxidants, proteins and sugars [48]. The transformations occurred mainly during baking but also during the resting period of the biscuit dough.

### 2.6. Pearson Correlation

In order to study the correlation between the % of spent malt rootlets addition and the volatile compounds/fatty acids of the final baked products, Pearson correlation was run. In the final baked goods, hexanal amount increased as the % of MRF increased having a total value of 28.22% in the biscuits enriched with 25% MRF. The amount of hexanal in MRF (8.36%), increased 3.37 times in the final baked goods with 25% MRF, having a Pearson correlation of 0.9558. The Pearson correlation between the amount of 3-methylbutan-1-ol from the SMR and the final baked goods is 0.991 which indicated a strong relationship between this raw material and the final products. Also, Limonene content of the MRF influenced the amount of the final baked goods, showing a strong relation by having a positive Pearson correlation of 0.9753. In contrast, from the fatty acids, only linoleic acid reached a Pearson correlation value of 0.8266, followed by myristoleic (0.6593), linolenic (0.5425), lauric and oleic acids with correlation values of 0.518 and 0.2435, respectively.

### 2.7. Sensory Analysis

The enriched biscuits showed significant differences for sensory descriptors compared to the control sample. The enriched biscuits were characterized by more intense color, flavor, taste, odor as the % of MRF increased. The biscuits with 15% MRF reached the highest hedonic score, having a mean sensory score for the visual appearance (color), taste and flavor of 8.1, 8.2 and 8.3 respectively, as showed in the Figure 3. Also, regarding the texture and overall acceptability, the panelists preferred the biscuits with 15% MRF, having a total score of 7.9 and 8.1 respectively. This was also confirmed by [13], who reported that an amount of 20% MRF in the biscuits caused the rejection of the product, due to the flavor, appearance and texture.

In the case of the lower color hedonic scores reached at 20% and 25% MRF addition (6.5, 6.0, respectively), the panelists described the biscuits as having a dark color. As the percentage of MRF increased, the color of the biscuits darkened. This is in agreement with [6] who reported that MRF had an impact on the crust color of the bread. Also, the Maillard and caramelization reactions influenced the final color of the biscuits, according to [49].

Regarding the biscuits taste and odor, panelists described an intense note of whisky, alcohol partly imputable to the increment of 3-methylbutan-1-ol with the MRF addition. Also, for the samples with more than 15% MRF, panelists indicated an unpleasant aldehydic taste imputable to the hexanal amount which was negative correlated to the bread aroma by consumers [33]. Citrus, pine, and mint notes were indicated by panelists as pleasant aromas for enriched biscuits and are positively correlated with limonene and *β*-pinene.

## 3. Materials and Methods

### 3.1. Procurement of Raw Materials

Whole rice flour, buckwheat flour, whole wheat flour, chia seeds flour and oat flakes flour were purchased from specialized stores in Cluj-Napoca, România. According to the producers, all the ingredients meet the highest quality standards. Malt extract and natural lemon aroma were purchased from SC. Ireks Pan SRL București, România. Barley dried malt rootlets were received from SC. Soufflet Agro SRL Buzau, România and were obtained from the autumn barley *Sebastian Ukraina* variety in 2016. Barley malt rootlets are the dried shoots and rootlets of sprouted grain in the brewing process and is considered a by-product [6,10,50]. All reagents were of analytical grade. Analytical reagents and chemicals were purchased from Sigma Aldrich (Louis, MO, USA).

### 3.2. Malt Rootlets Flour Preparation

The dried rootlets were passed through a mill feeder processed by grinding to fine flour (<300 μm) on a Grindomix (Model GM200, Haan, Germany) laboratory mill at 10,000 rot/min for 50 s, fitted with a 0.8 mm sieve and then homogenised by mixing. Barley malt rootlets flour (MRF) was subjected by supplemented in different proportion to prepare five different digestive biscuits.

### 3.3. Proximate Composition Analysis of the Flours and Biscuits

The chemical characteristics were carried out according to AACC Approved Methods [51] like moisture (44–15.02), lipids (30–25.01), ash (08–01.01), crude fibre (32–07.01) and protein were measured using the Kjeldahl method (46–11.02), nitrogen to protein conversion factor was 5.7. Total carbohydrate (%) content was calculated as the difference: 100 − (moisture + ash + proteins + lipids + crude fibres) method reported in a previous work [52], and also reported by [53].

### 3.4. Preparation of Dough and Baking Biscuits

The formulation of biscuit dough was summarized in Table 6. The multigrain mix consisted of whole meal rice flour, buckwheat flour, chia seeds flour and oat flakes flour in a ratio of: 40:30:20:10 (WRF:BF:CSF:OFF). The formulation used for biscuits manufacturing (Table 6) was calculated taking into consideration a basis formed by spelt whole wheat flour and a multigrain mix in the following ratios 40:60 (SWF:MG). Malt rootlets flour (MRF) at 5%, 10%, 15%, 20% and 25% supplementation levels, were incorporated into multigrain mix to produce composite flour, according to Table 6.

To obtain the control samples (B0), all dry ingredients (whole meal rice flour (WRF), buckwheat flour (BF), spelt whole wheat flour (SWF), chia seeds flour (CSF) and oat flakes flour (OFF)) were mixed together in a mixer (KitchenAid^®^ Precise Heat Mixing Bowl., Ohio, OH, USA), butter and malt extract were previously emulsified and then added by incorporation into the flour mix and then combined with the rest of the ingredients previously emulsified as well (baking powder, lemon juice, natural lemon aroma and salt) according to the recipe presented. The dough was mixed for 10 min to obtain a homogeneous consistency and then placed into the rest room at 6 °C, over a period of 30 min. The dough was then rolled out and 5 mm thick biscuits with a 60 mm diameter were formed and baked at 200 °C for 12 min. The biscuits enriched with the malt rootlets flour were codified as follows: B5, B10, B15, B20 and B25, according the amount of MRF used for supplementation. These were then stored in glass containers for two months, protected from light, and at a room temperature of 21 °C. The results of all the experiments are given as the average of replicates. Each replicate (biscuit) was obtained from separately prepared batches of dough. The analyses were carried out on fresh biscuits (on the day of baking). The procedure of its preparation was adapted from another similar study [53].

### 3.5. Determination of Fatty Acid Composition

#### 3.5.1. Total Lipid Determination

Lipids were extracted in petroleum ether using a Soxtest Raypa SX-6 MP (Barcelona, Spain) apparatus. Three g of crushed biscuits sample was introduced in cartridges and 50 mL petroleum was used for each sample extraction. Parameters were set as follows: temperature −75 °C, time of extraction −50 min. Samples were dried to constant weight. Total fat was expressed as percentage from the sample (% *w/w*).

#### 3.5.2. Fatty Acids Profile by GC-MS Analysis

Fatty acid methyl esters (FAMEs) from total lipids were analyzed using a Shimadzu GCMS-QP2010PLUS apparatus and an AT-WAX column (30 m, 0.32 mm i.d., 1 µm thickness), (Shimadzu Scientific Instruments, Kyoto, Japan) belonging to the Interdisciplinary Research Platform of Banat’s University of Agricultural Sciences and Veterinary Medicine “King Michael I of Romania. The solvent used for FAMEs was hexane, while the carrier gas (helium) was used at a flow rate of 1.00 mL/min and a linear velocity of 37.8 cm/s. Derivatization was performed for 1 h at 80 °C in an ultrasonic bath. An aliquot of 0.1 g sample was treated with 3 mL boron trifluoride methanol solution 20% (*w/v*). After cooling, 2.5 mL NaCl solution 10% (*w/v*) was added and methyl esters were extracted in 2 mL hexane, the organic layer being separated by centrifugation at 1006 G for 15 min using centrifuge Z36 HK (Hermle Labortechnik GmbH, Wehinge, Germany) for 15 min. The hexane solution (1 µL) was injected in apparatus. Samples were analyzed keeping the column initially at 140 °C for 10 min and then increasing the temperature with 7 °C/min up to 250 °C and maintaining at this temperature for 10 min (total run: 35.71 min). Split ratio was 1:10 and injection port temperature was set at 250 °C. The ion source and interface temperatures were 210 °C and 255 °C.

FAME peaks were identified using NIST05 library and quantified by area normalization method. Some compounds were unidentifiable due to the lack of authentic samples and library spectra of the corresponding molecules. The percentage of various lipidic compounds was determined by reporting the peak area corresponding to a specific compound to the total peak area (for all identified constituents) of chromatograms [54].

### 3.6. Extraction and Analysis of Volatile Compound

The analysis of volatile compounds was carried out on a GCMS QP-2010 (Shimadzu Scientific Instruments, Kyoto, Japan) gas chromatograph-mass spectrometer equipped with a CombiPAL AOC-5000 auto-sampler (Kyoto, Japan). The extraction of volatile compounds from samples was performed using the in-tube extraction technique (ITEX) [43] with some modification. Three grams of crushed biscuit samples were introduced into 20 mL headspace vial, incubated at 60 °C for 20 min under continuous stirring; the volatile compounds in the gas phase were repeatedly adsorbed (30 extractions) by fibre syringe ((ITEX-2TRAPTXTA, Tenax TA 80/100 mesh) and then were thermally desorbed directly into the GC-MS injector.

The volatile compounds were separated on a Zebron ZB-5MS (Phenomenex) capillary column of (30 m, 0.32 mm i.d., 1 µm thickness). The carrier gas was helium, the flow rate was 1 mL/min and the split ratio 1:5. The chromatographic column temperature program has been set by: 35 °C (held for 5 min) rising to 155 °C with 7 °C/min and then heated to 260 °C with 10 °C/min and held for 5 min. The injector, ion-source and interface temperatures were set at 250 °C. The MS mode was electron impact (EI) at ionization energy of 70 eV. The mass range scanned was 40–400 m/z. The volatile compounds were tentatively identified based on the spectra of reference compounds from NIST27 and NIST147 mass spectra libraries (with a similarity of minimum 85%) and verified by comparison with retention indices drawn from [55] or [56]. All peaks found in at least two of the three total ion chromatograms (TIC) were taken into account when calculating the total area of peaks (100%) and the relative areas of the volatile compounds [23,38].

### 3.7. Sensory Evaluation

The sensory characteristics of the biscuit sample was carried out by 60 panellists, 30% male and 70% female, range: 20–65 years. Each panellist analysed two biscuits from each sample, taking into account the texture, flavour, taste, overall acceptability and appearance on a nine-point Hedonic scale, in the following sequence: 1–4 represent negative sensations, 5 was neither like nor dislike, and 6–9, positive sensations, 9 meaning extremely like.

### 3.8. Statistical Analysis

The results of three independent (*n* = 3) assays performed with replicates each were expressed as means ± standard deviations. Data were compared using Duncan multiple comparison test by using SPSS version 19 software (*p* ≤ 0.005). Pearson correlation was also done to correlate the MRF percentage of addition with the volatile compounds content.

## 4. Conclusions

To the best of our knowledge, there are very few works reporting on the relationship between the spent malt rootlets addition in baked goods and their fatty acids, volatile and sensory profiles. Spent malt rootlets are by-products of the brewing industry and are a good source of valuable nutrients. However, their specific aroma profile could limit their addition in baked products. The results of this study reveal the fatty acids and volatile profile of spent malt rootlets. Moreover, the same profiles are determined for enriched biscuits with spent malt rootlets pointing out the contribution of fatty acids to the generation of aroma compounds. Also, a correlation between the addition of spent malt rootlets and the aroma and sensory profile of biscuits was done in order to obtain information about how different volatile compounds influence the consumer’s preferences and finally, what is the optimum percentage of addition. Future studies will report on the nutritional and quality parameters of multigrain biscuit enriched with spent malt rootlets.

## Figures and Tables

**Figure 1 molecules-25-00442-f001:**
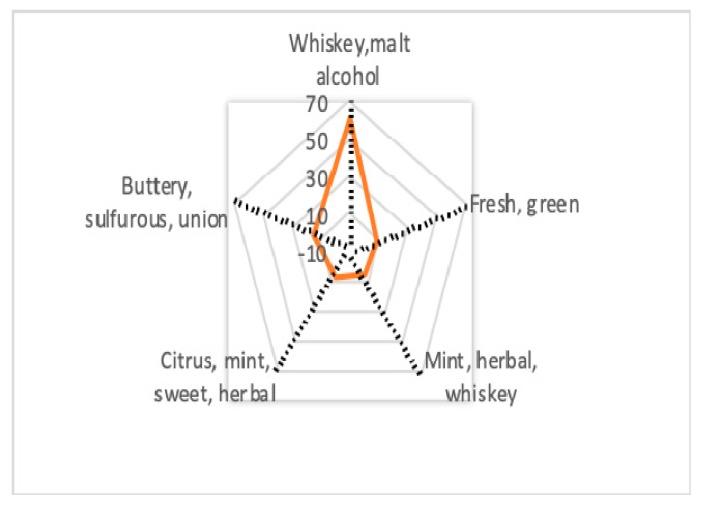
Aroma profile of spent malt rootlets by main compounds classes identified during GC-MS analysis.

**Figure 2 molecules-25-00442-f002:**
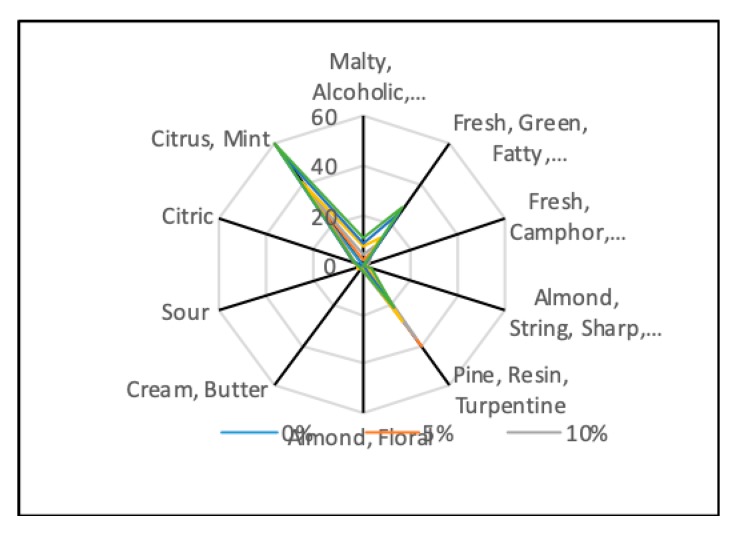
Aroma profile of enriched biscuits with spent malt rootlets in different concentrations (0–25%).

**Figure 3 molecules-25-00442-f003:**
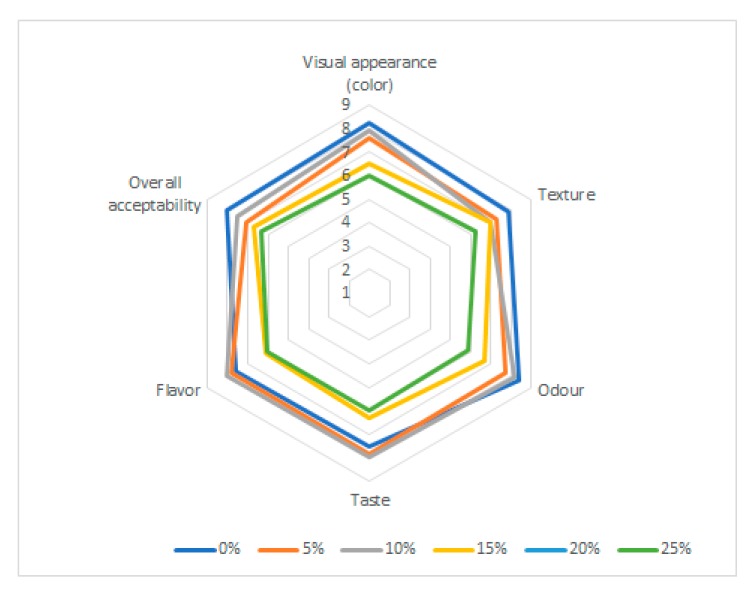
Hedonic scores for enriched biscuits with spent malt rootlets in different concentrations (0–25%).

**Table 1 molecules-25-00442-t001:** Proximate composition of spent malt rootlets (MRF), whole wheat spelt flour (SWF), buckwheat flour (BF), whole rice flour (WRF), oat flakes flour (OFF), and chia seeds flour (CF) used as raw materials in enriched biscuits formulation.

Parameters	MRF	SWF	BF	WRF	OFF	CF
Moisture	8.2 ± 0.6 ^b^	14.27 ± 0.3 ^e^	12.3 ± 0.2 ^d^	10.5 ± 0.4 ^c^	13.3 ± 0.3 ^de^	6.41 ± 0.1 ^a^
Proteins	35.5 ± 0.3 ^d^	15 ± 0.3 ^bc^	13.7± 0.2 ^b^	8.5 ± 0.1 ^a^	13.8 ± 0.2 ^b^	15.9 ±0.2 ^c^
Lipids	1.9 ^a^ ± 0.3 ^a^	2.9 ± 0.3 ^a^	2.8 ± 0.23 ^a^	2.5 ± 0.3 ^a^	6.1 ± 0.5 ^b^	29.3 ± 0.2 ^c^
Carbohydrates	46.70 ± 1.4 ^a^	66.59 ± 1.0 ^bc^	69.10 ± 0.21 ^bc^	77.6 ± 0.6 ^c^	63.70 ± 1.4 ^b^	46.19 ± 0.1 ^a^
Ash	7.7 ± 0.2 ^c^	1.24 ± 0.2 ^ab^	2.1 ± 0.3 ^ab^	0.9 ± 0.1 ^a^	3.1 ± 0.2 ^b^	2.2 ± 0.2 ^ab^

Each value was the mean of duplicate measurements; a–e Different superscripts in a row indicate significant difference within samples (*p* < 0.05).

**Table 2 molecules-25-00442-t002:** Comparative fatty acids methyl esters content (% of total fatty acids methyl esters) of spent malt rootlets flour (MRF) and whole wheat spelt flour (SWF).

Shorthand Nomenclature	Fatty Acids Systematic Names	Fatty Acids Trivial Name	Type	FA (% From Total FA) Samples	
				SWF	MRF
10:0	Decanoic acid	Capric acid	SFA	0.58 ± 0.21	0.31 ± 0.03
10:1(n-6)	*cis*-4-Decenoic acid	Obtusilic acid	MUFA, ω−6	0.05 ± 0.01	0.14 ± 0.04
12:0	Dodecanoic acid	Lauric acid	SFA	1.13 ± 0.21	0.69 ± 0.17
15:0	Pentadecanoic acid	Pentadecylic acid	SFA	0.19 ± 0.15	0.42 ± 0.16
a15:0	Methyl-tetradecanoic acid	Sarcinic acid	SFA	0.15 ± 0.03	nd
16:0	Hexadecanoic acid	Palmitic acid	SFA	23.23 ± 0.11	30.50 ± 0.36
16:1(n-7)	*cis*-9-Hexadecenoic acid	Palmitoleic acid	MUFA	0.69 ± 0.09	0.26 ± 0.08
17:0	Heptadecanoic acid	Margaric acid	SFA	3.24 ± 0.05	0.03 ± 0.01
18:0	Octadecanoic acid	Stearic acid	SFA	4.11 ± 0.02	1.45 ± 0.17
tr 18:1(n-9)	*trans*-9-Octadecenoic acid	Elaidic acid	MUFA, ω-9	nd	0.09 ± 0.01
18:1(n-7)	*cis*-11-Octadecenoic acid	Vaccenic acid	MUFA	nd	1.15 ± 0.29
18:1(n-9)	*cis*-9-Octadecenoic acid	Oleic acid	MUFA, ω-9	24.66 ± 0.04	12.13 ± 0.09
18:2(n-6)	9,12-Octadecadienoic acid	Linoleic acid	PUFA, ω−6	26.69 ± 0.03	35.61 ± 0.26
18:3(n-3)	9,12,15-Octadecatrienoic acid	Linolenic acid	PUFA, ω−3	4.42 ± 0.19	32.64 ± 0.38
20:0	Eicosanoic acid	Arachidic acid	SFA	0.11 ± 0.06	nd
20:4(n-6)	5,8,11,14-Eicosatetraenoic acid	Arachidonic	PUFA, ω−6	nd	0.79 ± 0.19
22:1(n-9)	*cis*-13-Docosenoic acid	Erucic acid	MUFA, ω-9	nd	0.38 ± 0.15
22:6(n-3)	4,7,10,13,16,19-Docosahexaenoic acid	Docosaheptaenoic acid	PUFA, ω−3	nd	1.16 ± 0.53
∑SFAs	-	-	-	32.63 ± 0.78	33.40 ± 0.90
∑MUFAs	-	-	-	25.40 ± 0.14	14.15 ± 0.66
∑PUFAs	-	-	-	31.11 ± 0.22	70.20 ± 1.36
∑PUFAs/∑MUFAs	-	-	-	1.22 ± 1.57	4.96 ± 2.06
∑PUFAs/∑SFAs	-	-	-	0.95 ± 0.28	2.10 ± 1.51

Note: Each value was the mean of triplicate measurements. FA-fatty acid, SFA-saturated fatty acids, PUFA-polyunsaturated, MUFA-monounsaturated fatty acids, nd-not detected; Each value was the mean of duplicate measurements.

**Table 3 molecules-25-00442-t003:** Volatile compounds of spent malt rootlets flour (MRF) and whole wheat spelt flour (SWF) ^*^.

Volatile Compounds	RI (Retention Indices)	Perceived Flavour	SWF (%)	MRF (%)
**Alcohols**				
3-methylbutan-1-ol	736	Whiskey, malt, burnt	-	40.21 ± 0.23
2-methylbutan-1-ol	739	Alcoholic, winey	-	4.02 ± 0.49
pentan-1-ol	759	Fruity, sweet	-	3.25 ± 0.16
hexan-1-ol	851	Ethereal, Oil, alcohol, green, Fruity, Sweet, Woody, Floral	-	12.72 ± 0.38
oct-1-en-3-ol	975	Mushroom, herbal, earthy	-	1.46 ± 0.23
**Aldehydes**				
Hexanal	801	Fresh, Green, Fatty, Aldehydic, Grass, Leafy, Fruity, Sweaty	35.96 ± 0.37	8.36 ± 0.51
Heptanal	903	Fresh, Aldehydic, Fatty, Green, Burgundy, Grass	-	0.88 ± 0.49
Nonanal	1106	Aldehydic, rose, waxy, citrus, orange, floral	6.80 ± 0.28	-
2-methylpropanal	-	Wine, malt	49.20 ± 0.47	-
**Ketone**				
heptan-2-one	895	Fruity spicy, sweet herbal, coconut woody	-	2.09 ± 0.39
octan-3-one	965	Herbal	-	2.94 ± 0.65
1-phenylethanone	1041	Almond, floral	8.02 ± 0.32	-
Ethanone	-	Whiskey	-	0.29 ± 0.21
2-methylpentan-3-one	-	Mint	-	0.25 ± 0.33
**Terpenes and terpenoids**				
*p*-Cymene	1028	Citrus, sweet, herbal, spicy	-	0.58 ± 0.66
Limonene	1031	Citrus, mint	-	6.08 ± 0.48
**Others**				
3-methylbutyl acetate	877	Sweaty, fruity, solvent	-	0.54 ± 0.27
Nonane	900	Solvent	-	0.68 ± 0.48
(E)-3,5,5-trimethylhex-2-ene	969	Solvent	-	2.47 ± 0.77
*2*-pentylfuran	993	Buttery, green beans	-	10.25 ± 0.11
(methyldisulfanyl) methane	-	Sulfurous, onion, cabbage	-	1.45 ± 0.24

Each value was the mean of duplicate measurements.

**Table 4 molecules-25-00442-t004:** Fatty acids methyl esters content (% of total fatty acids methyl esters) in enriched biscuits with spent malt rootlets in different concentrations.

Shorthand Nomenclature	Fatty Acid’s Name	Type	B0	B5	B10	B15	B20	B25
4:0	Butyric	SFA	1.75 ± 0.05 ^d^	1.11± 0.11 ^a^	1.19 ± 0.08 ^b^	1.51 ± 0.05 ^c^	1.77 ± 0.13 ^d^	1.81 ± 0.09 ^d^
6:0	Caproic	SFA	0.90 ± 0.09 ^a^	0.92 ± 0.17 ^a^	0.98 ± 0.33 ^a^	0.86 ± 0.06 ^a^	0.98 ± 0.01 ^a^	1.41 ± 0.09 ^b^
8:0	Caprylic	SFA	0.73 ± 0.23 ^b^	0.83 ± 0.04 ^b^	0.85 ± 0.23 ^b^	0.84 ± 0.02 ^b^	0.04 ± 0.18 ^a^	0.04 ± 0.08 ^a^
10:0	Capric	SFA	2.30 ± 0.17 ^a^	2.38 ± 0.02 ^a^	2.48± 0.09 ^a^	3.39 ± 0.24 ^b^	2.31 ± 0.09 ^a^	2.24 ± 0.05 ^a^
10:1	Decenoic	MUFA	0.24 ± 0.55 ^a^	0.22 ± 0.19 ^a^	0.22 ± 0.12 ^a^	0.38 ± 0.01 ^b^	0.25 ± 0.07 ^a^	0.24 ± 0.81 ^a^
12:0	Lauric	SFA	3.56 ± 0.22 ^a^	3.34 ± 0.27 ^a^	3.37± 0.04 ^a^	3.75 ± 0.06 ^a^	4.63 ± 0.04 ^b^	5.10 ± 0.04 ^c^
14:0	Myristic	SFA	11.06 ± 0.08 ^c^	11.05 ± 0.15 ^c^	11.04 ± 0.02 ^c^	11.04 ± 0.17 ^c^	11.02 ± 0.12 ^b^	10.95 ± 0.01 ^a^
14:1	Myristoleic	MUFA	1.19 ± 0.03 ^ab^	1.11 ± 0.11 ^a^	1.18 ± 0.23 ^ab^	1.24 ± 0.08 ^bc^	1.33 ± 0.22 ^c^	1.84 ± 0.21 ^d^
16:0	Palmitic	SFA	0.44 ± 0.02 ^c^	0.35 ± 0.05 ^a^	0.41 ± 0.11 ^b^	0.41 ± 0.23 ^b^	0.40 ± 0.03 ^b^	0.40 ± 0.25 ^b^
18:3(n-3)	Linolenic	PUFA ω-3	1.98 ± 0.20 ^a^	1.92 ± 0.05 ^a^	1.99 ± 0.08 _a_	2.38 ± 0.06 ^a^	2.11 ± 0.07 ^a^	4.60 ± 0.04 ^b^
18:1(n-9)	Oleic	MUFA, ω-9	1.26 ± 0.22 ^c^	1.01 ± 0.14 ^a^	1.18 ± 0.02 ^b^	1.17 ± 0.39 ^b^	1.15 ± 0.03 ^b^	1.18 ± 0.44 ^b^
18:2(n-6)	Linoleic	PUFA, ω−6	29.13 ± 0.04 ^a^	32.36 ± 0.03 ^b^	32.62 ± 0.01 ^b^	38.78 ± 0.06 ^c^	47.13 ± 0.03 ^d^	51.67 ± 0.07 ^e^
∑SFAs	-	-	20.74 ± 0.86 ^bc^	19.98 ± 0.81 ^a^	20.32 ± 0.90 ^ab^	21.80 ± 0.83 ^d^	21.15 ± 0.60^c^	21.95 ± 0.61^d^
∑MUFAs	-	-	2.69 ± 0.80 ^bc^	2.34 ± 0.74 ^a^	2.58 ± 0.09 ^b^	2.79 ± 0.48 ^c^	2.73 ± 0.32 ^bc^	3.26 ± 1.46 ^d^
∑PUFAs	-	-	31.11 ± 0,24 ^a^	34.28 ± 0.28 ^b^	34.61 ± 0.09 ^b^	41.16 ± 0.12 ^c^	49.24 ±0.10 ^d^	56.27 ± 0.11 ^e^
∑PUFAs/∑MUFAs	-	-	11.57 ± 0.30 ^a^	14.65 ± 0.38 ^c^	13.41 ± 0.35 ^b^	14.75 ± 0.25 ^c^	18.04 ± 0.31 ^d^	17.26 ± 0.08 ^d^
∑PUFAs/∑SFAs			1.5 ± 0.28 ^a^	1.72 ± 0.10 ^b^	1.70 ± 0.10 ^b^	1.89 ± 0.14 ^c^	2.33 ± 0.17 ^d^	2.56 ± 0.18 ^e^
∑MUFAs/∑SFAs			0.13 ± 0.93 ^b^	0.12 ± 0.54 ^a^	0.13 ± 0.29 ^b^	0.13 ± 0.58 ^b^	0.13 ± 0.53 ^b^	0.15 ± 2.39 ^c^
∑PUFAs + ∑MUFAs/∑SFAs			1.63 ± 1.21 ^a^	1.83 ± 0.64 ^b^	1.83 ± 0.39 ^b^	2.02 ± 0.72 ^c^	2.46 ± 0.70 ^d^	2.71 ± 2.57 ^e^

Note: FA-fatty acid, SFA-saturated fatty acids, PUFA-polyunsaturated, MUFA-monounsaturated fatty acids, biscuits samples with different amount of spent malt rootlets (0%,5%,10%,15%,20%,25%). ^a–e^ Different superscripts in a row indicate significant difference within samples (*p* < 0.05). Each value was the mean of duplicate measurements.

**Table 5 molecules-25-00442-t005:** Volatile compounds content of the enriched biscuits with spent malt rootlets.

Volatile Compounds	RI (Retention Indices)	Perceived Flavour	B0	B5	B10	B15	B20	B25
**Alcohols**								
3-methylbutan-1-ol	736	Malty, alcoholic whiskey	-	1.34 ± 0.03 ^a^	4.12 ± 0.09 ^b^	7.45 ± 0.71^c^	8.41± 0.02 ^d^	11.27 ± 0.01 ^e^
**Aldehyde**								
Hexanal	801	Fresh, Green, Fatty, Aldehydic, Grass, Leafy, Fruity, Sweaty	4.51 ± 0.06 ^a^	6.21 ± 0.25 ^b^	9.29 ± 0.28 ^c^	12.58 ± 0.29 ^d^	25.06 ± 0.11 ^e^	28.22 ± 0.26 ^f^
Benzaldehyde	960	Almond, String, Sharp, Sweet, Bitter, Cherry	07.1 ± 0.02 ^abc^	0.39 ± 0.55 ^a^	1.37 ± 0.07 ^bc^	1.47 ± 0.01 ^c^	0.6 ± 0.03 ^ab^	3.9 ± 0.36 ^d^
**Ketone**								
heptan-2-one	895	Fruity spicy, sweet herbal, coconut woody	0.88 ± 0.02 ^b^	-	-	-	0.51± 0.33 ^a^	-
1-phenylethanone	1042	Almond, floral	1.34 ± 0.07 ^b^	1.78 ± 0.02 ^bc^	2.49 ± 0.28 ^cd^	1.3 ± 0.56 ^b^	0.39 ± 0.44 ^a^	2.79 ± 0.08 ^d^
**Terpenes and terpenoids**								
α-Thujene	938	Green, herbal, woody	-	-	-	-	0.38 ± 0.55	-
*α*-Pinene	939	Fresh, Sweet, Green, Woody, Earthy	1.53 ± 0.05 ^a^	2.77 ± 0.24 ^e^	2.38 ± 0.45 ^d^	2.27 ± 0.22 ^c^	1.75 ± 0.22 ^b^	-
*β*-Pinene	982	pine, resin, turpentine	21.81 ± 0.66 ^a^	21.16 ± 0.21 ^a^	27.76 ± 0.77 ^b^	29.9 ± 0.12 ^c^	36.78 ± 0.09 ^d^	40.24 ± 0.03 ^e^
*β*-Myrcene	992	Tropical, fruity with mango shades, grassy	-	-	-	-	1.84 ± 0.21	-
*α*-Phellandrene	1006	Citrus, woody, grassy, pepper	-	-	-	1.19 ± 0.06 ^a^	2.15 ± 0.44 ^b^	-
*p*-Cymene	1028	Citrus, sweet, herbal, spicy	2.98 ± 0.31 ^a^	3.01 ± 0.23 ^a^	4.17 ± 0.44 ^b^	4.53 ± 0.03 ^b^	4.68 ± 0.04 ^b^	2.54 ± 0.22 ^a^
Limonene	1031	Citrus, mint	24.76 ± 0.38 ^a^	33.41 ± 0.56 ^b^	35.22 ± 0.09 ^b^	41.49 ± 0.37 ^c^	55.91 ± 0.03 ^d^	60.15 ± 0.45 ^e^
*γ*-Terpinene	1074	Citrus, tropical, fruity, oily, woody	-	-	-	-	3.02 ±0.05	-
1,3,8*-p*-Menthatriene	1110	Woody, citrus, grassy	-	4.39 ± 0.05	-	-	-	-
**Acids**								
Benzoic acid	1277	balsamic	1.14	-	-	-	-	-
**Other compounds**								
2-pentylfuran	993	Butter, green beans	-	-	-	-	-	3.32 ± 0.33
n.i.	-		1.54 ± 0.22 ^a^	2.01 ± 0.11 ^b^	1.41± 0.46 ^a^	1.98 ± 0.55 ^b^	-	1.79 ± 0.41 ^b^
n.i.	-		2.09 ± 0.44 ^bc^	2.33 ± 0.51 ^c^	2.13 ± 0.33 ^c^	1.64 ± 0.38 ^b^	0.81 ± 0.88 ^a^	2.16 ± 0.22 ^c^

Note: ^a–d^ different superscripts in a row indicate significant difference within samples (*p* < 0.05). Each value was the mean of duplicate measurements.

**Table 6 molecules-25-00442-t006:** Formulation used for the preparation of enriched biscuits.

Components (g)	B0 (Control)	B5	B10	B15	B20	B25
Spelt whole wheat flour (SWF)	40	40	40	40	40	40
Multigrain mix (MG)	60	55	50	45	40	35
Malt rootlets flour (MRF)	0	5	10	15	20	25
Butter	40	40	40	40	40	40
Malt extract	27	27	27	27	27	27
Baking powder	0.5	0.5	0.5	0.5	0.5	0.5
Lemon juice	1	1	1	1	1	1
Natural lemon aroma	0.5	0.5	0.5	0.5	0.5	0.5
Salt	0.2	0.2	0.2	0.2	0.2	0.2

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
