# Peer review of "Fatty Acids, Volatile and Sensory Profile of Multigrain Biscuits Enriched with Spent Malt Rootles"

_molecules, 2020, doi:10.3390/molecules25030442_

Round 1

Reviewer 1 Report

The authors studied the effect of spent malt rootlets addition on volatile derivatives of enriched biscuits in relation to their sensory profile. The analysis of the volatiles was performed using GC-MS technique. In chapter 3.6. the authors state verification by comparison with retention indices drawn from pherobase and flavornet, while similarity of mass spectra was stated to be of minimum 85%.  However, the main concern of the study is identification of the compounds, i.e. tables do not contain any information that can ascertain the identification such as retention indices, and should be presented in all the Tables.

Other specific remarks:

Table 2:  „cis“ should be writtten in italic

Table 3: nomenclature should be corrected (according to IUPAC!): 3-methylbutanol; 2-methylbutanol; pentanol (or pentan-1-ol); hexanol (or hexan-1-ol); 2-methylpropanal, heptan-2-one, 2-methylpentan-3-one, p-cymene;  limonene, 2-pentylfuran, 3,5,5-trimethylhex-2ene

Table 5: please correct nomenclature!

Page 11, Table 1?

Author Response

The responses for the reviewer are attached below.

Reviewer 2 Report

The manuscript describes the fatty acids, volatile and sensory profile of multigrain biscuits after enrichment with spent malt rootlets. The novelty of the manuscript is clear and the topic is worth of investigation. This reviewer supports the publication of this research paper, however some minor changes are necessary according to the following comments:

Line 71: Instead of [14,15,16,17], please write [14-17].

Lines 81-82: The sentence "Further papers....rootlets"should be deleted or moved to the conclusions.

Table 1: Please check the alignment in first row.

Line 125: Delete one space between and & palmitic.

The title of table 2 should be moved to the same page with the table. Moreover, at table 2 please write "arachidonic" instead of "arahidonic".

Line 138: please add a space at "notdetected"

Please move the title of table 3 and the context of the table in the page in order not to split it. Please do the same for table 4 and 5. Furthermore, at table 4 some numbers have comma instead of full stops e.g. "0,987" and "47.,131" for B20. In table 5 check the font style of the first row.

Please move the legend of figure 2 in the page with the figure.

Line 251: Instead of "0,8266" please write "0.8266".

Line 271: Instead of "3-Methyl-1-butanol" please write "3-methyl-1-butanol".

Section 3.1: Please provide the city and country for all companies.

Section 3.4: Please change the table 1 to table 6.

Line 284: Please change the [13],[7], [49] appropriately.

Line 285: Please add a fullstop at the and of the sentence.

Line 310 and 314: please delete + symbol and add the oC.

Line 321: Please use 3.5.1 for the subtitle "Total lipid determination". Please do the same for line 327.

Line 329: Please add a space at 1 μm. 

Line 331: Please change "hour" with "h". 

Line 332 : Rephrase sentence starting with number.

Line 333: Define % for NaCl concentration (w/w or w/v).

Line 334: Provide centrifugation speed in "g" instead of rpm. Also change "minutes" with "min".

Line 337 and 349: Please change "min." with "min".

Line 338: Please add a space before the first sentence.

Line 348: Please change "g" with "grams".

Line 349: Please change "ml" with "mL".

Line 353:Please change "ms" with "MS". For the dimensions of the column please describe it as in line 329. Please provide information for the company of both columns.

Line 354: please say "the flow rate was 1 mL min-1".

Line 376: Please replace the phrase "From our knowledge" with "To the best of our knowledge".

It is important to recheck all the decimal digits and use only statistically significant figures of the numbers in all tables and throughout text. 

Some titles (e.g. 3.6, 3.7 and 3.8) have fullstops at the end of the title. Please remove it and check throughout text.

Be consisted in using "/" or "-1" in units and check it throughout text.

Moreover, the manuscript needs further language revision to avoid some grammatical errors.

Author Response

The response for the reviewer are attached below.

Reviewer 3 Report

The Manuscript by Maria Simona ChiÅŸ and co-authors (Molecules-686370) entitled “ Fatty Acids, Volatile and Sensory Profile of Multigrain Biscuits Enriched with Spent Malt Rootles” provides some novel and interesting information on nutritional composition of spent malt rootles and their potential use in food industry.

In my opinion, this manuscript should be accepted for publication with some further corrections:

My comments:

In Materials and methods, triplicates are stated as have been used for calculations of the means and StDev. In the tables, duplicates are mentioned? StDev are shown only in the table 1, but they can not be calculated based on duplicates. Adjust the decimals - no need to provide three decimals –two will be accurate according the sensitivity on the method used. Provide StDev for all the tables/values. Table 2: 15:00 should be 15:0; 20:00 should be 20:0; adjust 10:0 in the table; no need to include 24:1 if it has not been identified; StDev needed as for all other tables; be consistent with the decimals. Please, explain why the ratio of PUFA/MUFA was used but not PUFA/SFA? Table 4: are C4:0 and C6:0 not volatile? How were they analysed and calculated? C18:2 – 47., 131 should be 47.131. Line 218/219: mammals because of the lack of endogenous enzymes for omega-3 desaturation [31] – there are two essential fatty acids: C18:2n6 and C18:3n3; two enzymes delta 12 and delta 15 are lacking in animals; correct this sentence. Correct the References with a required letter case.

Author Response

(The authors gave the same response as above.)

Round 2

Reviewer 1 Report

There are still correction required regarding nomenclature, e.g.:

D-limonene - "D" represents relative configuration that can not be determined by the column used. There is also an "L" configuration and it represents enantiomer of this one. Please erase "D" in table 3. and throughout the text! alpha, beta, para etc. should be italic (e.g. β-Pinene; p-Cymene) - this should also be corrected in the tables and throughout the text 2-pentylfuran (Table 2)

After these minor corrections manuscript can be accepted for publication.

Author Response

The response to the minor revisions are attached below.
